# Electrochemical cells from water ice? Preliminary methods and results

**Daniel S. Helman** [1,2]*, Matthew Retallack[3]

**1** Sustainability Education, Prescott College, Prescott, Arizona, United States of America, **2** Education Division, College of Micronesia-FSM, Colonia, Yap, Federated States of Micronesia, **3** School of Public Policy and Administration, Carleton University, Ottawa, Ontario, Canada

* dhelman@comfsm.fm, danielhelmanteaching@yahoo.com

**Data Availability Statement:** The authors have reserved a DOI on figshare for the data from this research project. DOI: 10.6084/m9.figshare. 16641034.

**Funding:** The authors received no specific funding for this work.

## Abstract

Electrochemical cells from ice will be an important seasonal addition to power generation in cold regions. We demonstrate power generation on the order of 0.1 mW at 0.3 V and 0.13 m² surface area using an electrochemical cell with 2% HCl providing a pH gradient in ice, and suggest a solar add-on effect due to temperature changes under direct sunlight. Different models are discussed, and data are presented related to different additives: (1) solutes such as NaCl and monopotassium phosphate; (2) pH modifying agents such as acids and bases; (3) particulate suspensions with kaolinite and other substances. The results are positive and suggest viable use of electrochemical cells from ice with low fabrication costs and safe environmental impact for ephemeral power generation, especially with future material improvements and refinement of technique. Current research in this nascent field is also briefly introduced. The model presented has implications both for power systems and for biology: an icy-worlds hypothesis for the origin of life suggests a protometabolism with an ice-based pH gradient.

## Introduction

This paper describes an early proof-of-concept stage in the development of photovoltaic cells from water ice using photoelectrochemical cells. The present paper is the first step, i.e. electrochemical cells from water ice. The abundance of water and the ease with which ice can be worked make it an attractive choice for photovoltaic energy production in places where ambient temperatures are cold. Likewise, the proof-of-concept herein can be used to develop modest functioning electrochemical cells from water ice for use in cold climates.

The present study was undertaken as field work. There was no clear model established yet. That arose in the failures and successes of the first experimental season, and inspired the experimental model that was followed during the second season with the experiment described in this paper. Real-world conditions and the various failures described herein and in the Supporting Information (S1 File) allowed for wildly disparate hypotheses to be worked on until a practical path became clear. Likewise, descriptions of technical problems and other errors that were eliminated during the present work are meant to help future research and to motivate more precise, controlled study in a laboratory setting as a follow-up.

**Competing interests:** The authors have declared that no competing interests exist.

It is important to consider that the conductance of electric charge in ice is not as a semiconductor whose basis is electron movement. Band gap measurement, for example, which is so critical in determining material choices in typical photovoltaic development [1], is not the critical parameter in determining free motion of charge in ice. Doublet (D-type) Bjerrum defects predominate over empty (L-type) defects in the conduction process of ice [2], and this two-proton structure highlights how a protonic mobility can occur readily and be mediated by pH. See Fig 1.

As with other solid electrochemical cells, the electronegativity of ions present in the structure of the ice helps to establish electromotive pathways and a voltage across the device. The monatomic ions studied in this experiment are $K^+$, $Na^+$, $Cl^-$ (with electronegativities: 0.82, 0.93, 3.16, respectively) and come from NaCl, HCl and $KH_2PO_4$ additives. $[H_2PO_4]^-$ has a dissociation coefficient of 7.20 and can both donate and receive hydrogen ions forming $[HPO_4]^{2-}$ (dissociation coefficient 2.14) or $H_3PO_4$, phosphoric acid (dissociation coefficient 12.37).

At ambient terrestrial conditions, the electronic structure of ice is paraelectric, and this effect may heighten (or dampen) a voltage from being established as there will be structural

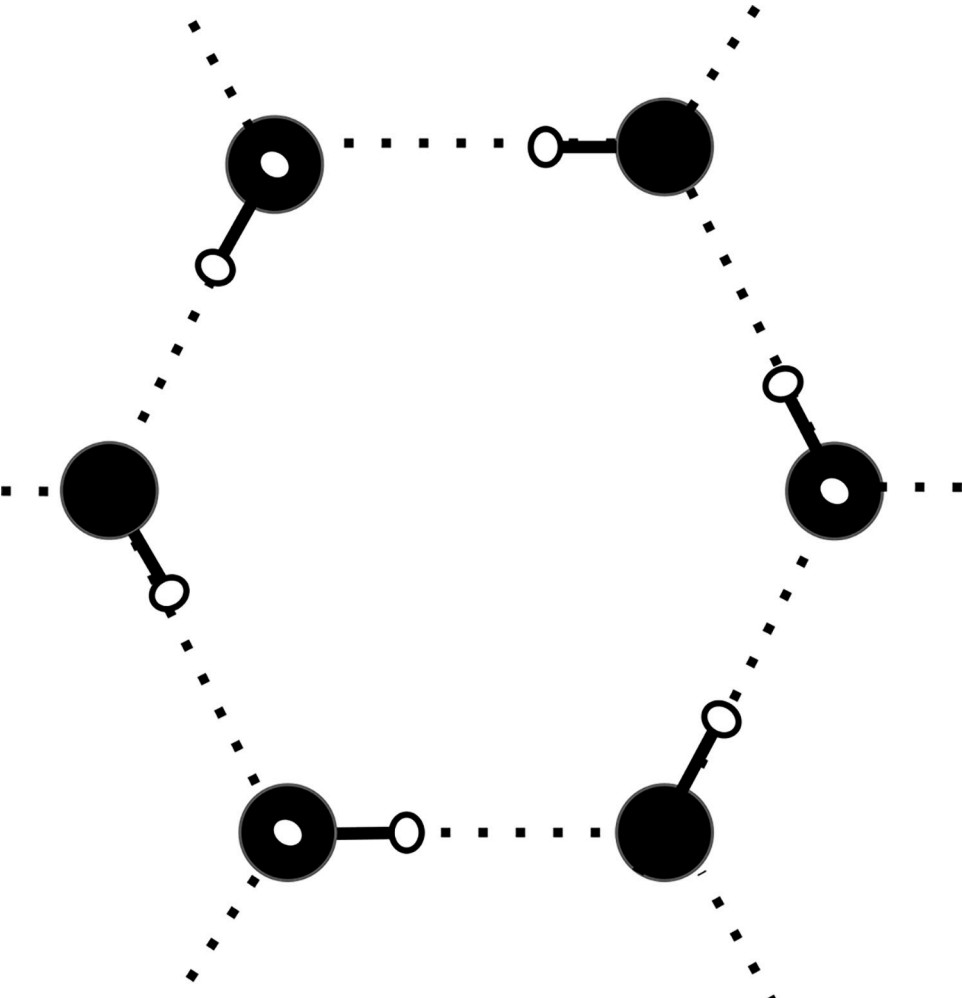

**Fig 1. Idealized single basal plane of ice Ih (Common water ice).** Oxygen atoms without hydrogen coming up out of the plane of the page have them going down to connect with the next lower basal layer, with hydrogen bonds normally formed with a single proton throughout. Bjerrum defects are either double proton (D-type) or empty of protons (L-type).

rearrangement of the material in response to the voltaic field. The effect is similar to that of a ferroelectric material, but with less ordering.

The electronic structure of the crystal lattice for common water ice also has the possibility of affecting charge accumulation, but this effect is small compared to ionic conductivity. For example, water ice is both pyroelectric and piezoelectric, and ice expresses $2 \times 10^{-12}$ C N$^{-1}$ of charge along the c-axis when deformed along c-axis, i.e. $d_{33}$, at 0˚C [3]. This is similar in magnitude to other piezoelectric crystals found in nature [4]. Notwithstanding, the paraelectric, pyroelectric and piezoelectric properties of ice are not well quantified. Data are sparse.

The electric conductivity of ice lies between that of conductors and insulators, with values on the order of $10^{-4}$ to $10^{-8}$ siemens per meter—and this is a large range for DC conductivity, more than three orders of magnitude. Moreover, electric conductivity in ice is frequency dependent, implying that the phenomenon is not based on a simple electron transfer mechanism. Petrenko [2] states that the presence of opposing sublattice orientations is responsible for its different conductivity modes.

Electric conduction in ice is related to the propagation of hydrogen bond defects (i.e. with two protons bonded to the electron, a D-type Bjerrum defect) within the crystal lattice aligned as hydronium ions ($H_3O^+$). Conduction pathways arise wherein one neighbor influences the state of an adjacent neighbor. The most mobile charge carrier is the proton and not the electron. Thus, while not a conductor nor insulator, ice is also not a semiconductor in the same fashion as silicon. Notwithstanding, the presence of other ions and defect types can promote conduction.

The Bjerrum defect effect and ionic effect in ice are the subject of additional ongoing studies and are well established. In the few years since this research was carried out, advances have been made in developing technology using similar principles, e.g. proton batteries [5] and hydronium ion batteries [6], and fundamental work has been done quantifying ice as a solid electrolyte conducting various ions [7]. In Guo et al. [7], fast freezing was employed to crystallize a saturated aqueous solution of sulfate salts in different trials with conductivities ranging from $10^{-7}$ S cm$^{-1}$ ($Zn^{2+}$) to $10^{-3}$ S cm$^{-1}$ ($Li^+$).

Fig 2 shows a model schematic of the prototype, i.e. electrochemical cells with an intermediate acid or base layer between the two ice layers. An idealized model is shown in Table 1 along with the cell thicknesses. Here, the theoretical mechanism is based on the motion of protons H + and hydronium $H_2O^+$ in the middle layer, with additives in solution and suspension in the top and bottom layers, designed to promote coordinated motion of charge carriers, with positive going one direction and negative the other. The top layer should have anions (–) or metals or other sources of electrons in abundance to attract the protons. The bottom layer should have cations (+) or oxidizers in abundance to attract electrons. Two cells of identical composition were used, typically with one elevated and exposed to face the Sun, and the other placed low, in shade and covered by plywood. Error in thickness is ±0.3 cm.

Details of the experiment are given in the section that follows. Descriptions of additional experiments, including solid-solid trial runs, can be found in the Supporting Information (S1 File) showing experimental plans. detailed results from two experimental seasons, and a description of problems that arose and how these were addressed.

## Materials and methods

Experiments took place in Ottawa, Canada at 45.412640˚ N, 75.704380˚ W from 2016 to 2017, and used ambient cold-weather conditions. Temperature and voltage data were taken at 1-minute intervals. Resistance measurements were taken iteratively. In addition to the data collected from each trial, 1-minute geomagnetic data were downloaded from the World Data

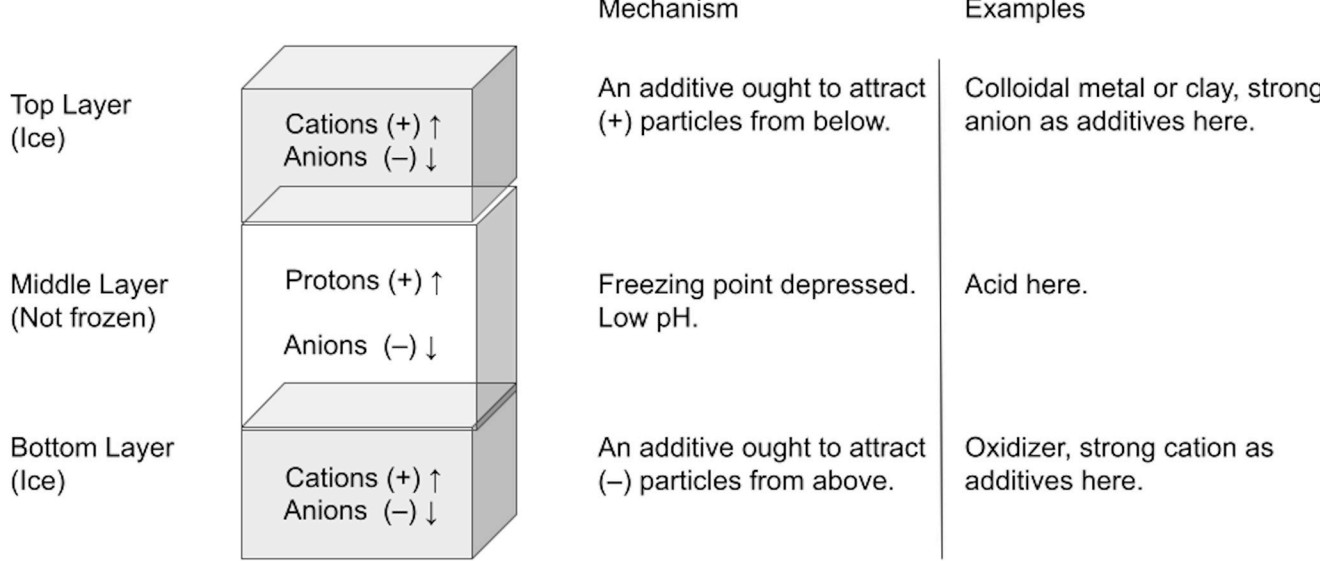

**Fig 2. General model for planning an electrochemical cell from water ice.** Acid is shown. A base model has charges reversed.

Center for Geomagnetism from the Ottawa (OTT) observatory (44.597° N, 75.552° W) (http://www.wdc.bgs.ac.uk/dataportal/). Lunisolar data, e.g. sunrise times, were downloaded from Edwards Apps (http://www.sunrisesunset.com/calendar.asp). Dawn and Dusk data are civil, i.e. with the Sun 6° below the horizon.

The ice cells were made from steam distilled water, purchased from a Canadian business (Water Mart). Restaurant bus pans (43.2 cm x 30.5 cm x 11.5 cm; surface area 0.13 m$^2$) from HDPE were used to house the cells.

The top electrodes were high transparency stainless steel woven 50 mesh 0.0012" wire diameter (TWP Inc) cut to fit. The mesh allows for sunlight to penetrate. The mesh was not perfectly even, e.g. there was a 0.5 cm diameter hole in one of the electrodes. Bottom electrodes were 0.032" thick bare sheet aluminum 1100-H14 (Online Metals Insc) cut to size 24" x 24". Long leads of copper wire were soldered onto the electrodes using standard lead-tin solder.

Additives were sourced as follows: Monopotassium phosphate (MKP) ($KH_2PO_4$) was purchased from Canadawide Scientific, Ltd. Powdered kaolinite clay, was from an art supplier. NaCl and muriatic acid were sourced as household items locally. Glass cylinders (tea light holders) to use as spacers in the middle layer were sourced from a home goods store.

The middle layer comprised 4.5 liters of distilled water plus acid. The dilution of muriatic acid was as follows: 286ml of HCl (31.45%) was added to 4.5 liters of distilled water to get a 2% HCl solution.

Cell construction consisted of mixing the distilled water with the solute and/or particles for suspension. Wooden frames were used to keep the top electrode in place during the freezing process. The electrodes sat approximately 0.6 cm below the surface of the top layer, as shown

**Table 1. Theoretical mechanism for the charge characteristics of the prototype cell.**

|  | Layer Composition | Mechanism | Height (cm) |
|---|---|---|---|
| **Top:** | $H_2O$ (2.5 L) + kaolinite (50 g) + NaCl (50 g) | $Cl^-$(↓) kaolinite $e^-$charging (↓) | 2.5 |
| **Middle:** | $H_2O$ + muriatic acid (HCl) (2%) | $H^+$(↑) $Cl^-$(↓) | 3.2 |
| **Bottom:** | $H_2O$ (2.5 L) + MKP ($KH_2PO_4$) (100 g) | $K^+$(↑) $H^+$(↑) | 2.5 |

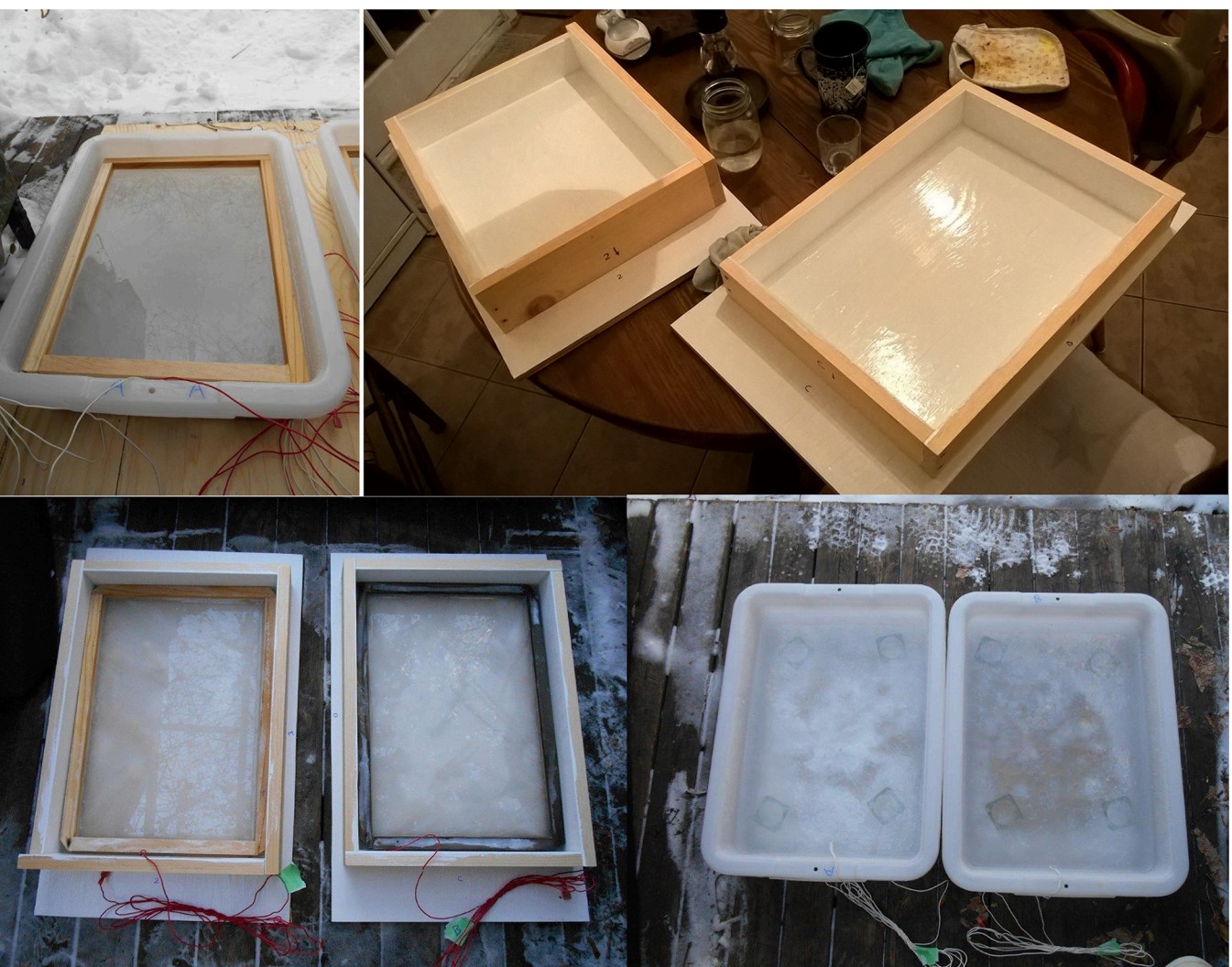

**Fig 3. Cell construction.** Top row from left: (A) Prototype cell from water ice showing the top electrode affixed to a wooden frame. (B) Wooden forms for fabricating the top cells. Forms were sanded and painted (high-gloss, two coats) inside to promote removal. Bottom row from left: (C) Freezing the top cell layers in wooden forms, for subsequent removal and transfer to the bus pans. The top electrodes in their wooden frames are also visible. (D) Middle layer preparation with glass spacers visible.

in Fig 3A. Cells were covered with a sheet of cardboard for the night and allowed to freeze. The construction of multi-layered cells required that individual layers be frozen independently and then assembled. Fig 3B and 3C shows the construction of the top layer. Fig 3D shows the middle layer after the bottom layer had set. Table 2 lists the basic steps of fabrication for these electrochemical cells.

Cells were arranged so that one was elevated and exposed, and the other was placed in shade and covered with plywood, and both were laying flat. The arrangement was switched during part of each trial, so that each cell had equal time in each position for an experimental control.

Ambient temperature was measured with a K-Type thermocouple attached to a UTC-USB hub (Omega Engineering) connected to a notebook computer for logging using the native Omega software app. See Fig 4A. The specified temperature range for K-Type thermocouples is −200˚ to 1250˚C. The standard error listed by the manufacturer is the greater of 2.2˚C or

**Table 2. Protocol for the construction of electrochemical cells from ice.**

| Step | Protocol |
|---|---|
| 1 | Construct a form for freezing the top layer, to be removed later. |
| 2 | Freeze the bottom layer in the bus pan and the top layer in the constructed form. See Fig 3(B). |
| 3 | Start the temperature log. |
| 4 | Once the bottom layer is frozen, add the intermediate slush layer with glass spacers. See Fig 3(D). |
| 5 | After the slush layer is suitably frozen, remove and install the top layer. |
| 6 | Begin logging voltage data. |

0.75%. The thermocouple hub was attached near a doorframe with the thermocouple probe set through the margin of the door to the ambient air outside, as shown in Fig 4B. Temperature data were taken at 1-minute intervals. Data were compared with ambient conditions (e.g. sunrise, shade, etc) and inspected for obvious inaccuracies and drift in temperature, and none were identified.

Voltage was measured with handheld UT61 digital multimeters (DMM) manufactured by Uni-T, connected via RS232 to USB converter to a notebook computer. Voltage data were taken at 1-minute intervals. In addition, a resistance box (Elenco RS 500 Resistance Substitution Box) with resistors of known value was connected in parallel to each DMM separately at set times, e.g. before the regular trial, voltage was measured several times and a resistance box was set iteratively to increasing resistances (1 $\Omega$, 5 $\Omega$, 10 $\Omega$, 100 $\Omega$, 1k $\Omega$, 100k $\Omega$, 1M $\Omega$) while voltages were logged, so that a range of currents could be calculated, and from these a range of power outputs were then calculated. See Fig 5. During the regular trials, voltages were measured with the 1k $\Omega$ resistor connected in parallel and power time series and energy output were calculated. Energy was calculated from the sum of the square of the voltages, i.e. E = $\Sigma$

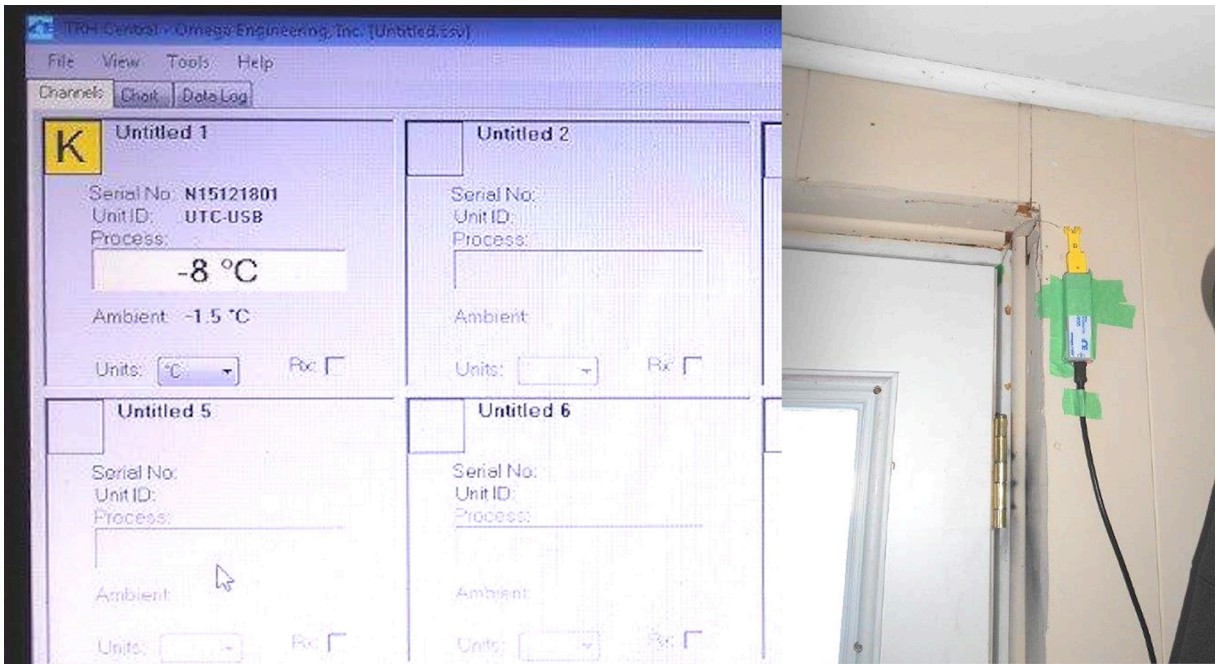

**Fig 4. Thermocouple setup for measuring temperature.** (A) Data set up screen. (B) Thermocouple hub showing K-type thermocouple probe inserted into the gap between door and frame leading to the outside.

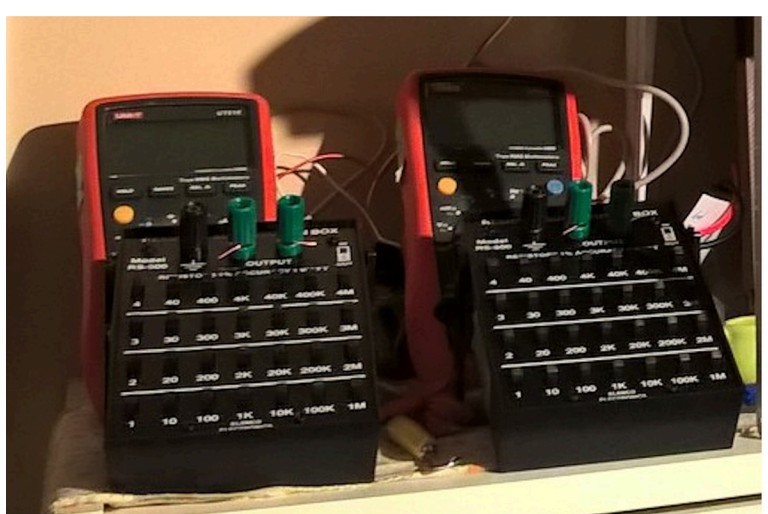
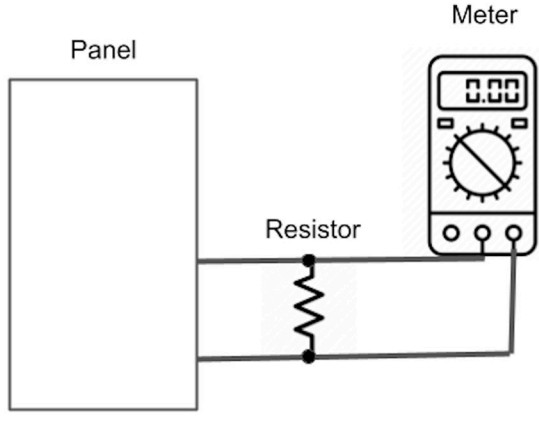

**Fig 5. Power and voltage measurement.** (A) The two variable resistance boxes for use in measuring power. The DMMs are visible in the background. (B) Diagram showing resistance measurement circuit of a prototype cell, with resistor connected in parallel. (Image: adapted from MTM Scientific. http://www.mtmscientific.com/solarpanel.html).

($V^2/I$), and then, as this is one-minute data, divided by 60 to calculate Wh. Data logging was in practice sparser than 1 per minute, e.g. 1 per 62 seconds, so a correction factor based on expected/actual points was introduced to correct the calculations.

A final note about lead-tin solder and the copper wires used to connect to the cell electrodes: Thermoelectric effects from the dissimilar metals used for the wires connecting with solder to the bottom and top electrodes can create small currents that may confound results if the target units are small. Lead tin solder may contribute a maximum thermoelectric effect of 5 $\mu V \degree C^{-1}$ [8] and will influence the measurement if the power output is low enough.

## Results

The experiment consisted of two trial cells each with 1 kΩ resistors attached in parallel. Table 3 summarizes the voltage, current, power and energy data gathered, and presents data from each trial when the cells were covered and uncovered separately, and then for each trial in total.

For the (kaolinite+NaCl)-HCl-MKP cells, median power was 0.1 mW for Trials 1 and 2. Over the course of 9 days, the Trial 1 cell produced 20 mWh of electricity. Much of the Trial 2 data were lost due to two computer faults, but for the 4.5 days of recorded operation it produced 10 mWh of electricity. Roughly, each cell produced 2 mWh of electricity per day at 0.3 V.

Standardized resistors were replaced successively three and four times, respectively during each trial, to measure the performance characteristics of the cells. The power and energy output varied depending on the value of the resistor, as shown in Fig 6. Each set of measurements

**Table 3. Voltage, current and power data with 1 kΩ load.**

| Trial | $V_{mean}$ | sd | $V_{median}$ | $I_{median}$ | $P_{median}$ | $P_{mean}$ | $E_{total}$ | Comments |
|---|---|---|---|---|---|---|---|---|
| 1 covered | 0.214 V | 0.090 V | 0.2548 V | 0.255 mA | 64.9 μW | 53.8 μW | 5.36 mWh | Noisy |
| 1 exposed | 0.362 V | 0.079 V | 0.4135 V | 0.414 mA | 171 μW | 137 μW | 16.1 mWh | |
| 1 all | 0.294 V | 0.112 V | 0.2610 V | 0.261 mA | 68.1 μW | 98.7 μW | 21.4 mWh | Noisy |
| 2 exposed | 0.315 V | 0.024 V | 0.3250 V | 0.325 mA | 106 μW | 100 μW | 2.34 mWh | Noisy |
| 2 covered | 0.291 V | 0.058 V | 0.2760 V | 0.276 mA | 76.2 μW | 87.9 μW | 7.22 mWh | Noisy |
| 2 all | 0.296 V | 0.053 V | 0.2984 V | 0.298 mA | 89.0 μW | 90.6 μW | 9.56 mWh | Noisy |

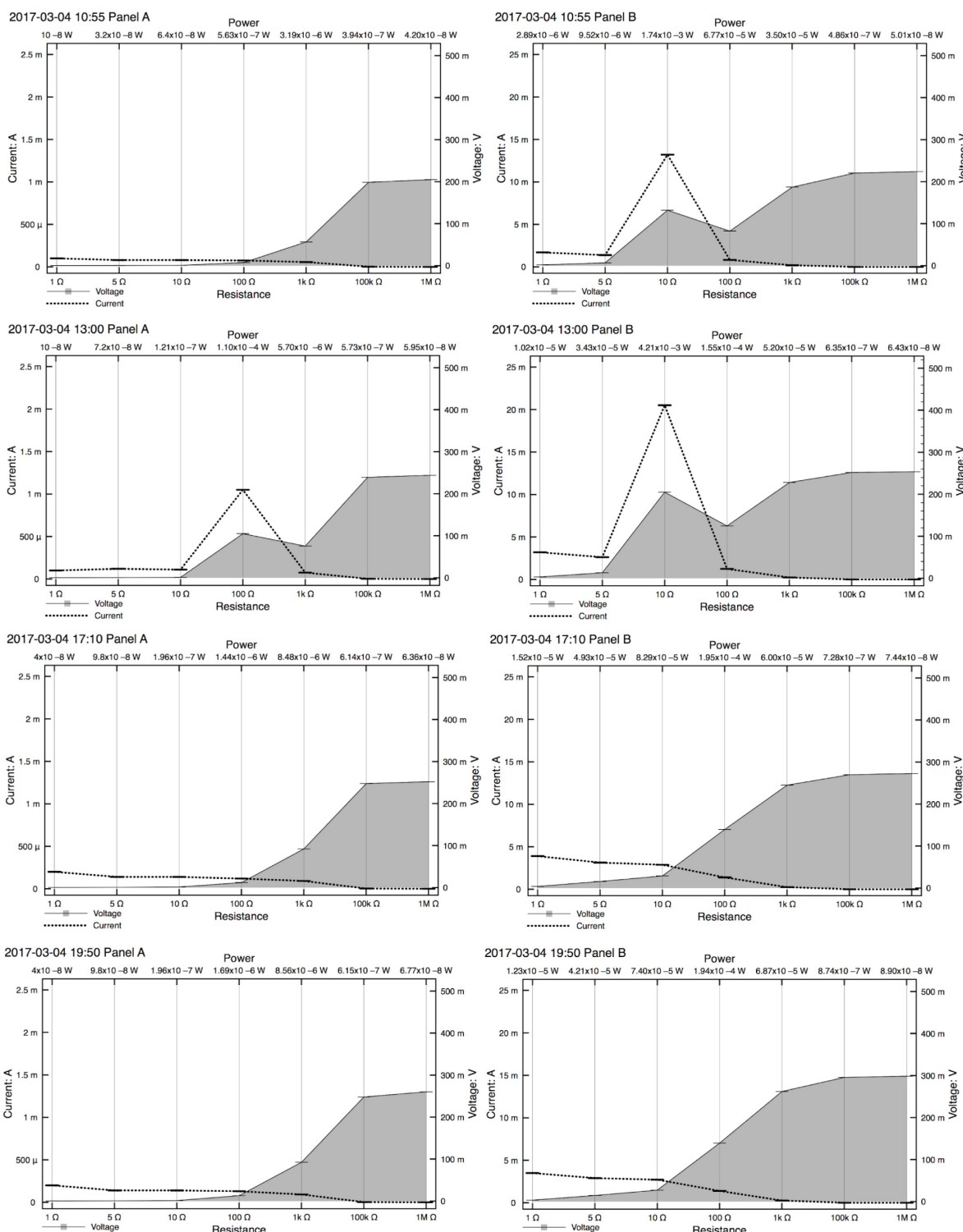

**Fig 6. Voltage, current and power.** Measured voltage and calculated current and power for prototype cells from (kaolinite+NaCl)-HCl-MKP, with resistors of known value connected in parallel. Orders of magnitude for the power values are listed in-line for ease of reading but should be read as superscripts, e.g. the top left power value of the top left cell should be read as $10^{-8}$ W. The shaded plot on the right of each graph shows power production. Panel A refers to the electrochemical cell of Trial 1; Panel B refers to the electrochemical cell of Trial 2. During these measurements Panel A was covered and Panel B was exposed. Each set of measurements took approximately 15 minutes to complete. Current scale (amperes) for Panel B is 10x the scale for Panel A.

**Table 4. Resistance associated with maximal voltage, current and power.**

|  | Resistor $t_1$ | Resistor $t_2$ | Resistor $t_3$ | Resistor $t_4$ |
|---|---|---|---|---|
| $V_1$ max | 1 MΩ | 1 MΩ | 1 MΩ | 1 MΩ |
| $V_2$ max | 1 MΩ | 1 MΩ | 1 MΩ | 1 MΩ |
| $I_1$ max | 1 Ω | 100 Ω | 1 Ω | 1 Ω |
| $I_2$ max | 10 Ω | 10 Ω | 1 Ω | 1 Ω |
| $P_1$ max | 1 kΩ | 100 Ω | 1 kΩ | 1 kΩ |
| $P_2$ max | 10 Ω | 10 Ω | 100 Ω | 100 Ω |

Note: Subscripts for voltage ($V$), current ($I$) and power ($P$) refer to different trials, and these ran concurrently during each experiment. Subscripts for the time ($t$) refer to each of three or four times that different values of resistors were successively replaced in each circuit and measurements taken. Resistance values were taken four times.

took approximately 15 minutes to complete. Table 4 shows the resistor value that produced voltage, current and power maxima during these measurements. One sees that these resistances are the same or vary by at most one order of magnitude within each cell for each of voltage, current and power except one. The 100Ω maxima for current and power for Trial 1 all occurred during a transient period of direct sun when the cell was covered.

Ice formation is a dynamic process and changes in form during the trials were common. These included surface bulging, surface pitting or irregular surface texture, loss of transparency, exsolution of solute as a surface film whether as a crust or as a liquid. There were also ice nucleations preferential to the wires of the device and walls, plus regions of entrapped fluid and bubbles of fluid, as well as spatially organized bubble nucleations. Rainfall during the experiment affected the surface of one of the cells (Trial 2) and led to inhomogeneities. Descriptions of these changes to the material of the cells as well as complete descriptions of this experiment and several additional experiments are included in the Supporting Information (S1 File).

## Discussion

Ice is a protonic semiconductor, and the most mobile charge carrier is the proton. The pH of the system is the major design consideration. Power production for the cells from both trials was ongoing for more than nine days without exhaustion of the output at 0.1 mW at 0.3 V for a cell surface area of 0.13 m$^2$ and this is consistent with the electrochemical model presented in the introduction. A 2% HCl solution for the middle layer, and NaCl and kaolinite for the top, and MKP for the bottom layer promote electromotive potential via pH and ionic considerations.

Clay particles like kaolinite tend to maintain a negative charge in aqueous environments, and kaolinite in particular has a zeta potential influenced by pH, on the order of –25 to –42 mV [9]. In the trials, kaolinite settled to the interface with the middle layer. Its presence is consistent with electromotive pairing to H$^+$ or H$_2$O$^+$ from the middle. Excess Cl$^-$ from NaCl in the top layer provides a concentration gradient promoting Cl$^-$ in the middle from HCl traveling down where it may pair with H$^+$ or H$_3$O$^+$ from ice at the bottom layer interface. MKP in the bottom layer also provides an additional source of H$^+$ as well as K$^+$ ions to act as a target for Cl$^-$ pairing electromotively at the bottom layer interface. Here is a short recap of what we found. During the trials, exposed and covered cells both increased power production during periods of direct sunlight; and, overall, exposed cells produced more power.

The frequency dependence in ice conduction, i.e. during alternating current experiments found in the literature [3], is due to the shape and position of the sublattices of the water ice crystal. As with other ferroelectric and paraelectric crystals, different frequencies allow for

sublattice excitations that either constructively or destructively influence the output. In water ice, the conduction here is due to hydrogen bonds, and thus to hydronium transfer and protonic conduction. Bjerrum defects, whether related to the presence or absence of a proton where one is expected in the crystal lattice influence hydrogen bond conduction. If there is a photonic response to water ice, it may be instantiated in hydrogen bonds.

Thus, in addition to ionic influences on pH and the effect of the Bjerrum defects to allow for local increases in conduction, it is also possible that the interaction of incident photons with these defects in an exposed cell may produce a secondary source of electrical output, working in conjunction with protonic conduction, as, typically, defects change band gap structure [10]. Our experiments did not find evidence to support or rule out this hypothesized ice photoelectric effect. Recall that higher power production occurred when cells were exposed, and also when covered during heating.

During all of the experiments, there were many instances of changes to the form of the cells and this suggests recrystallization effects. The strongest fit with the evidence is that a solar effect may have been caused by recrystallization of the ice, and includes contributions both from energy of fusion and energy of dissolution as solutes on freezing tend to be segregated out of the crystal lattice [11]. The additional solar energy may allow remanent solutes within a crystal to be expelled to the margins. Since recrystallization is a thermodynamic process, the process includes activation energy and a transition to stability, and this will also include production of electrical signals in a dynamic process as crystallization and electricity interact [12]. The power production optimization seen at middle resistances is consistent with this result. Ice systems are dynamic [13]. At times they correlate with the solar influx or temperature [14]. At times a peak in temperature will initiate an interval of structural reorganization, e.g. Kuroda and Lacmann [15], and this is seen as changes in voltage.

### Hypotheses and future development

In addition to temperature or solar effects, another idea is that the interaction of the geomagnetic field with the ice may have affected power output. Sunrise and sunset influence telluric currents and the local field. See Supporting Information (S1 File) for some examples. However, there are no visual correlations between total geomagnetic field strength and the voltage data gathered. Thus one may rule out electromagnetic induction in the ice (even in the presence of the paraelectric effect) as a primary source of electrical activity during these trials. Had excursions in measured voltage or power in the prototypes produced a correlation with the total geomagnetic field strength, there would be additional questions about an active mechanism. See Table 5 for a brief description of this and additional mechanisms to influence electrical behavior. These may form the motivation for controlled laboratory experiments in the future.

The discussion that follows includes relevant data (where known) to elucidate the mechanisms listed in Table 5. Bjerrum defects may be important as incident photons interact with the electronic structure of the material and the hydrogen bonds [16]. Li and Ross [17] found two molecular optic bands in inelastic neutron spectra for ice I solid phase, implying two different strengths of hydrogen bond, one strong and one weak. That is, a change in the Bjerrum defects may change the band gap of water ice (7.8 eV) via changes to the length, orientation and abundance of the hydrogen bonds. A photoelectric effect involving hydrogen-bond amplification may occur to promote power production. Notably, the photoelectric effect in minerals is implicated in the development of photosynthesis in cyanobacteria [18] and a similar mechanism may be present here.

Semiconductor additives which can stay suspended, such as kaolinite, with a band gap of 4.52 eV [19], also may be mixed with water to form ice. Likewise, silicon, a common

**Table 5. Known and hypothesized electrical mechanisms for future work.**

| Mechanism | Description |
|---|---|
| Bjerrum Defect Effect | Bjerrum defects promote proton mobility. |
| Ionic Effect | The presence and type of ions promotes conduction. |
| Other Defect Effect (hypothesis) | The presence and type of other defects promotes conduction. |
| Ice Photoelectric Effect (hypothesis) | Incident sunlight changes the electron band gap and promotes conduction. |
| Additive Photoelectric Effect (hypothesis) | Incident sunlight with photoelectric additives such as kaolinite promotes conduction. |
| Thermodynamic Effect (hypothesis) | Phase change coupled with solute partitioning promote conduction. |
| Geomagnetic Effect (hypothesis) | Geomagnetic field induces an electric field which promotes conduction. |
| Pyroelectric-Piezoelectric Effect (hypothesis) | Changes due to changes in temperature, and the expansion or contraction of the ice crystal lattice produces electrical charge. |

semiconductor used in photovoltaic applications, has a band gap of 1.2 eV. A photoelectric effect may occur here to promote power production. That is, electronic charge builds up to promote a voltage differential.

Additives and solutes segregate during ice formation according to respective partition coefficients which can be experimentally determined, and which result in differential concentrations in the liquid and solid-liquid interface [20]. The partition coefficient K is the ratio of the concentration ($C$) of the solute in the solid and liquid phases,

$$K = C_{\mathrm{S}}/C_{\mathrm{L}} \tag{1}$$

and influences the thermodynamics of the ice system via changes due to heat of dissolution/precipitation of the solute. For n solutes, Eq 1 is repeated for each solute.

$$K_{\mathrm{n}} \in \mathbf{K} : \ \{K_1, K_2, \ldots K_{\mathrm{n}}\} \tag{2}$$

Thus, the standard heat equation for the precipitation of ice,

$$Q = m_0 L_{\mathrm{F}} \tag{3}$$

Where $Q$ is the heat, $m_0$ is the mass of ice and $L_{\mathrm{F}}$ is the latent heat of fusion, is modified according to

$$Q = m_0 L_{\mathrm{F}} + \mathbf{mKL_D} \tag{4}$$

where

$$m_{\mathrm{n}} \in \mathbf{m} : \ \{m_1, m_2, \ldots, m_{\mathrm{n}}\} \tag{5}$$

and

$$L_{\mathrm{Dn}} \in \mathbf{L_D} : \ \{L_{\mathrm{D1}}, L_{\mathrm{D2}}, \ldots, L_{\mathrm{Dn}}\} \tag{6}$$

with $m_{\mathrm{n}}$ and $K_{\mathrm{n}}$ and $L_{\mathrm{Dn}}$ representing the mass, partition coefficient and latent heat of dissolution, respectively, for each solute $n$ in the ice system. Thus, this segregation process and the thermodynamics of freezing, thawing and freeze-thaw cycles will affect the amount of energy available for the proton mobility in our device. This is the thermodynamic effect listed in Table 5, and Eqs (1–6) will be useful for its study.

As noted in the introduction, paraelectric, pyroelectric and piezoelectric effects are observable in water ice. A paraelectric effect caused by local changes to the geomagnetic field will

result in induced changes to the electric field of the ice. As with other paraelectric materials, the resulting field changes will be nonlinear and are related to the competing sublattice interactions. The effect may be less than what is measurable at these scales. And in fact, the experimental evidence suggests that this effect is negligible.

Likewise with the pyroelectric and piezoelectric effects. The pyroelectric effect occurs as charge accumulates at the distal ends of a polar crystal as the temperature changes, and is distinct from structural deformation to the crystal lattice due to temperature changes. That structural deformation is a piezoelectric effect.

It will be useful to clarify several points from these experiments for which the experimental setup for this proof-of-concept work was inadequate. Laboratory experiments with precise control of parameters, such as temperature, ice quality, duration and intensity of illumination, will be appropriate to provide a stronger quantification of these preliminary results. Several hypotheses listed in Table 5 remain unresolved. First, how critical is the variability of temperature, i.e. the "freezing regime", in terms of the internal structure and operation of the cells? Structural differences in the ice may change power outcomes. Likewise, one wonders whether the apparent solar effects seen on 4 March 2017 at 13:00 was due to temperature changes locally from direct sunlight on the covered cell, and whether kaolinite or a thaw-freeze cycle had an impact on that process.

Moreover, there is a new research community here, a growing body of research to explore proton batteries [5], hydronium ion batteries [6], and research into ice as a solid electrolyte conducting various ions [7], as well as new development in proton electrodes [21–23] and other related technologies, e.g. anolyte chemistry for seawater batteries [24].

There is also an additional consideration related to economy and practice. All ionic compounds as additives used for this and the experiments listed in the Supporting Information (S1 File) in particular were chosen for safety and availability. In general, the safest ionic compounds to work with are those that are common in nature geologically and generally regarded as safe: sodium chloride, calcium/magnesium sulfate, and sodium bicarbonate.

One idea presently is for further design to be community-led. There are large areas of open space that lie fallow in the wintertime in many high-latitude regions, and many large areas that are completely unused. In urban settings, considerations of weight for load-bearing structures (on rooftops and exposed decks) may prevent some experiments but allow others. An interested party can readily run experimental trials and further develop the technology, while also being considerate of the ecology. Natural systems like glaciers and frozen lake or sea ice may be an important focus of primary research in that proper electrode placement may harvest some natural electrical output.

Likewise, additional controlled experiments are warranted, as this proof-of-concept study is consistent with making electrochemical batteries from ice for use in cold climates, so long as optimization can lead to greater power output by two or more orders of magnitude. Likewise, a photosensitive material additive which influences pH, and can be added to a top layer may allow for photovoltaic cells from ice, again for use in cold climates. With the addition of proper protonic electrodes, this work may be useful.

## Biological considerations

A proton pump is a protein-based structure that aids the transfer of protons across a membrane in living systems. It allows for a pH gradient across the membrane that can be used for biological processes. There are a number of different mechanisms/types of proton pumps, and these have evolved several times within living organisms independently [25]. Modest production of electricity via the action of pH gradients in ice may be important for studies about the

origin of life in icy worlds, as they provide a pathway for abiogenesis. This work may form the basis for further study related to protometabolic processes, and especially regular, cyclical systems such as tidal force piezoelectricity, and also autocatalytic systems in ice generally. For a more detailed background description of these processes, see Supporting Information (S1 File).

## Conclusion

Fabrication of ice cells for energy production is now in a very early stage of development. This paper demonstrates the ease with which to do primary research, and includes positive results, i.e. 0.1 mW at 0.3 V for each of two 0.13 $m^2$ cells for nine days of operation. Several challenges lie ahead. One of these is insuring that fabrication will continue to focus on safe materials. It would be tragic if agricultural soils were spoiled, for example, with large amounts of acid used to generate a few Wh of electricity. Additional work also needs to be done to characterize the behavior of ice plus solutes and suspended particles in a cell in order to elucidate the action of direct sun, and to characterize under which conditions electrical production matches closely to temperature, and by which mechanism that activity occurs, and when electrical output follows other mechanisms. Similarly, understanding which electrode materials are economic to use in conjunction with ice cells is another important area of need. This work presents the electrochemical cell as a successful model for power production from ice, with a well-planned flow of protons and ions. The most mobile charge carrier in water ice is the proton, and pH gradients in ice are notable both for modest energy production and as a basis for understanding a hypothesized origin of life on icy worlds.

## Supporting information

**S1 File. Sections include additional details, figures and tables related to experimental plans and methods, detailed descriptions of results, experimental failures, and also geomagnetic correlations, as well as a longer treatment of astrobiology and the origin of life as they relate to this research.**
(DOC)

## Acknowledgments

The project arose from a presentation by the first author (DH) at the European Consortium for Political Research (ECPR) General Assembly in Montreal, Canada in 2015 entitled "Ice as a Novel Material for Solar Panels in Cold Climates: Potential for Adoption and Usage" while enrolled as a doctoral student in Sustainability Education at Prescott College. Fieldwork was undertaken by the second author (MR) in a household setting in Ottawa, Canada, with the support and keen interest of the author's family. Data plots were made using Plot2 for Mac (version 2.6.17) which is a free app designed and maintained by Michael Wesemann (https://apps.micw.org/).

## Author Contributions

**Conceptualization:** Daniel S. Helman.

**Data curation:** Daniel S. Helman, Matthew Retallack.

**Formal analysis:** Daniel S. Helman.

**Investigation:** Matthew Retallack.

**Methodology:** Daniel S. Helman, Matthew Retallack.

**Project administration:** Daniel S. Helman, Matthew Retallack.

**Resources:** Daniel S. Helman, Matthew Retallack.

**Software:** Matthew Retallack.

**Visualization:** Daniel S. Helman.

**Writing – original draft:** Daniel S. Helman.

**Writing – review & editing:** Daniel S. Helman, Matthew Retallack.

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
