## [Decision Letter · Decision Letter 0]

21 Feb 2023

PONE-D-23-02888Electrochemical cells from water ice? Preliminary methods and resultsPLOS ONE

Dear Dr. Helman,

Thank you for submitting your manuscript to PLOS ONE. After careful consideration, we feel that it has merit but does not fully meet PLOS ONE’s publication criteria as it currently stands. Therefore, we invite you to submit a revised version of the manuscript that addresses the points raised during the review process.

We look forward to receiving your revised manuscript.

Kind regards,

Nasser A. M. Barakat

Academic Editor

PLOS ONE

Journal Requirements:

Reviewers' comments:

Reviewer's Responses to Questions

**Comments to the Author**

1. Is the manuscript technically sound, and do the data support the conclusions?

Reviewer #1: Partly

Reviewer #2: Partly

2. Has the statistical analysis been performed appropriately and rigorously? 

Reviewer #1: N/A

Reviewer #2: Yes

3. Have the authors made all data underlying the findings in their manuscript fully available?

Reviewer #1: Yes

Reviewer #2: Yes

4. Is the manuscript presented in an intelligible fashion and written in standard English?

Reviewer #1: Yes

Reviewer #2: No

5. Review Comments to the Author

Reviewer #1: The authors presented a somewhat unconventional manuscript for me, which deals with the very interesting idea of using ice as an very environment-friendly electrochemical battery. This idea is supported by some theoretical works that are cited. Based on this, the authors prepared several long-term experiments in real conditions to verify several hypotheses. It must be said that conducting experiments in real conditions with many variables is very difficult to prove the concept. The experiments were also accompanied by technical problems and other errors that were eliminated during the experiments. So it is rather a work that shows the way, what to watch out for when preparing such experiments. Even so, this work contains some interesting findings that are worth publishing.

However, the authors presented several hypotheses that could not be reliably refuted or confirmed, mainly due to the variation of many variables in real natural conditions. Laboratory experiments with precise control of parameters (such as temperature, ice quality, duration and intensity of illumination, etc.) would perhaps be more appropriate to verify the entire concept. The manuscript is written in a comprehensible way, although it is a bit difficult to follow all the experiments that were carried out, which were not taken into account, etc. It might be worth focusing mainly on the most successful experiment and describing it in more detail.

It is very difficult for me to judge this manuscript because I have no serious objection to how the data is presented. And I also understand the changes in the experiments that have been done over time. Most of the mistakes and shortcomings in the experiments are acknowledged and described, which can be a significant help in conducting other similar experiments.

Although the authors actually present several rather unsuccessful experiments and individual hypotheses could not be confirmed or refuted in this arrangement, I would be in favor of publishing these data.

Perhaps it would be worth, in the end, to design and describe what the future experiment should look like - what should be the appropriate composition and arrangement of panels and other structural elements? What conditions are critical (temperature, precipitation, ice transparency, etc.)? How to set up an experiment to assess the critical effect of temperature?

Reviewer #2: The manuscript reports an interesting study on the production electrical power using ice. The manuscript does not seem suitable for publication in its present form. Several hypothesis are stated in the introduction section, but the conclusion section does not provide any conclusive inference regarding these. The Results and Discussion sections are difficult to comprehend without first reading the Experimental section, which is given at the end of the manuscript. Unless they are rewritten for clarity, the standard approach of presenting the methods first, before discussing the results would be beneficial.

The introduction should be more focused and brief. Equations (1) to (6) are presented, but are not used anywhere in the analysis. If the conclusions of the study seem to indicate that the ionic effect is the most likely reason for the observed electrochemical effects, other effects (Bjerrum defect, ice photoelectric, additive photoelectric, thermodynamic, geomagnetic, pyroelectric, piezoelectric, etc.) must be presented in a shortened, simplified, and clearer form in the introduction. Also, it is clear early on in the manuscript that the ice being studied is frozen water. It is not necessary to state “water ice” in the later sections of the manuscript.

The introduction should also explain why the fieldwork was undertaken in favor of a more controlled study in a laboratory setting.

The Results section must also be greatly simplified. First, present experiments that worked, and the results and conclusions from these experiments clearly. Experiments that did not work should all be moved to Supplemental Information (section S3). The reviewer does not see the benefit of including the failed experiments in the main article. They make the key results difficult to comprehend.

There are numerous tables and figures in the present version. Only representative data must be shown in the main article, and the rest moved to Supplemental Information. The ordinate axis label for the power values in all the figures must be corrected, using the correct scientific format for numerical values.

While it is acknowledged that the authors are reporting only a preliminary study, a more detailed investigation of the electrical (ionic) conductivities of ice that contains the different additives would be desirable. I recommend that the authors present a much more concise and to-the-point report of their study (consistent with the preliminary nature of the investigation) in a revised version of the manuscript.

6. PLOS authors have the option to publish the peer review history of their article (what does this mean?). If published, this will include your full peer review and any attached files.

Reviewer #1: No

Reviewer #2: No

---

## [Author Response · Author response to Decision Letter 0]

14 Apr 2023

Electrochemical Cells from Water Ice? Preliminary methods and results

Response to Reviewers

28 March to 12 April 2023

The comments and suggestions of the two reviewers were very helpful in improving the manuscript, and the time taken is greatly appreciated. The introduction has been shortened, all but the most successful experiment have been moved to the Supporting Information file, the hypotheses have all been moved to the discussion section for motivation of future work. A rationale for doing this as field work (rather than laboratory investigations) has been added to the introduction as well. The main point that field work was preferred since there wasn’t a clear model established yet has been emphasized. A clear experimental model arose in the failures and successes of the first experimental season, and inspired the successes during the second season.

Two suggestions of the reviewers were not taken up, and I offer this short explanation.

• The conductivities of ice with the various additives in the successful experiment are not included, as this is a follow up (future stage) of this research. It will require laboratory work to gather the data. The preliminary work of this manuscript is in the successful experimental model described herein that was developed during these two field seasons.

• The details of specific future experiments to address each hypothesis in a laboratory setting are not included, as these can differ greatly, according to the interests of the researcher and equipment available, but need not differ substantially from what is presented in the manuscript. For example, transmission electron microscope (TEM) data can show freezing regime and the structure of where different ions are present in the experiment described.

To summarize: These are both good suggestions. For the first, it will be a future endeavor. For the second, the range of possible experiments is not well constrained, nor does it need to be. It depends on the interests and equipment available. I hope this brief explanation is acceptable to the editor and reviewers, and that the gratitude of the authors is evident, notwithstanding our not taking up these two good suggestions at this time.

The specific comments of each reviewer is listed below, along with a response to each. In some cases the reviewers’ comments were broken up to allow for responses to each part. The original order of the comments was retained. Comments by Reviewer #1 are listed first. Responses by the authors are indicated with “Author:”. Thank you!

Reviewer #1: The authors presented a somewhat unconventional manuscript for me, which deals with the very interesting idea of using ice as an very environment-friendly electrochemical battery. This idea is supported by some theoretical works that are cited. Based on this, the authors prepared several long-term experiments in real conditions to verify several hypotheses.

Authors: Thank you for your time and insight, and for your kind focus.

Reviewer #1: It must be said that conducting experiments in real conditions with many variables is very difficult to prove the concept. The experiments were also accompanied by technical problems and other errors that were eliminated during the experiments. So it is rather a work that shows the way, what to watch out for when preparing such experiments. Even so, this work contains some interesting findings that are worth publishing.

Authors: Thank you so very much. The following sentence was added after the first paragraph of the introduction to describe the rationale more clearly of the real-world experimental design.

“The present study was undertaken as field work. There was no clear model established yet. That arose in the failures and successes of the first experimental season, and inspired the experimental model that was followed during the second season with the experiment described in this paper. Real-world conditions and the various failures described herein and in the Supporting Information (S1 File) allowed for wildly disparate hypotheses to be worked on until a practical path became clear. Likewise, descriptions of technical problems and other errors that were eliminated during the present work are meant to help future research and to motivate more precise, controlled study in a laboratory setting as a follow-up.”

Reviewer #1: However, the authors presented several hypotheses that could not be reliably refuted or confirmed, mainly due to the variation of many variables in real natural conditions. Laboratory experiments with precise control of parameters (such as temperature, ice quality, duration and intensity of illumination, etc.) would perhaps be more appropriate to verify the entire concept.

Authors: Thank you so very much. This is a cogent insight, and all the hypotheses have been moved to the discussion section for clarity, as they can be the subject of future investigations.

Likewise, the following text has been added to the discussion section for clarity:

“Laboratory experiments with precise control of parameters, such as temperature, ice quality, duration and intensity of illumination, will be appropriate to provide a stronger quantification of these preliminary results. Several hypotheses listed in Table 5 remain unresolved.”

Reviewer #1: The manuscript is written in a comprehensible way, although it is a bit difficult to follow all the experiments that were carried out, which were not taken into account, etc. It might be worth focusing mainly on the most successful experiment and describing it in more detail.

Authors: Thank you for this good suggestion. Only the most successful experiment (Experiment 6) has been retained in the main text, for clarity. Details of the other experiments have been moved to the Supporting Information file. This change simplifies the text and makes it easier to follow.

Reviewer #1: It is very difficult for me to judge this manuscript because I have no serious objection to how the data is presented. And I also understand the changes in the experiments that have been done over time. Most of the mistakes and shortcomings in the experiments are acknowledged and described, which can be a significant help in conducting other similar experiments.

Although the authors actually present several rather unsuccessful experiments and individual hypotheses could not be confirmed or refuted in this arrangement, I would be in favor of publishing these data.

Perhaps it would be worth, in the end, to design and describe what the future experiment should look like - what should be the appropriate composition and arrangement of panels and other structural elements? What conditions are critical (temperature, precipitation, ice transparency, etc.)? How to set up an experiment to assess the critical effect of temperature?

Authors: Thank you for this excellent suggestion. The discussion section has been rewritten to focus more clearly on the hypotheses and what future experiments can try to resolve. Note that in general, a controlled laboratory experimental plan can follow compositional and other parameters described in the manuscript, and these should be successful, so long as there is access to equipment like a TEM, etc. in a cryogenic lab. Thank you so very much for your time and insight!

Reviewer #2: The manuscript reports an interesting study on the production electrical power using ice. The manuscript does not seem suitable for publication in its present form. Several hypothesis are stated in the introduction section, but the conclusion section does not provide any conclusive inference regarding these. 

Authors: The hypotheses have been moved from the introduction to the discussion section as a focus for future work, as is appropriate, since quantifying these effects is more closely related to laboratory experiments.

Reviewer #2: The Results and Discussion sections are difficult to comprehend without first reading the Experimental section, which is given at the end of the manuscript. Unless they are rewritten for clarity, the standard approach of presenting the methods first, before discussing the results would be beneficial.

Authors: Thank you for this good suggestion. The Experimental section has been moved forward to a place before the Results and Discussion section and has been renamed Methods.

Reviewer #2: The introduction should be more focused and brief. 

Authors: Thank you for this good suggestion. All of the hypotheses previously in the introduction have been moved to the discussion section, to allow for more clarity. Likewise, descriptions of the ionic effect and the various other electrical properties have been rewritten for brevity and with a focus more closely on the ionic effect.

Reviewer #2: Equations (1) to (6) are presented, but are not used anywhere in the analysis.

Authors: Thank you for pointing this out. For clarity, these equations have been moved to the discussion section where they are more appropriate for understanding how quantification in future studies will be done.

Reviewer #2: If the conclusions of the study seem to indicate that the ionic effect is the most likely reason for the observed electrochemical effects, other effects (Bjerrum defect, ice photoelectric, additive photoelectric, thermodynamic, geomagnetic, pyroelectric, piezoelectric, etc.) must be presented in a shortened, simplified, and clearer form in the introduction. 

Authors: Thank you for the good suggestion. For clarity, all of the hypotheses related to these effects were moved to the discussion section in order to make the introduction more focused. In their present form in the discussion, they can form the basis for future work.

Likewise, the following text was removed from the introduction in order to simplify the descriptions of the other effects which may be important for future studies but have less effect than the ionic:

“...wherein there are two competing crystal sublattices of different alignment,…”

“...than a ferroelectric, and this produces a non-linear (paraelectric) response to an applied electric field. The polarity of the crystal will not switch, as with a ferroelectric, as the sublattices interact with the electromotive force of the field, but there will be a structural reorganization.”

“...is also polar, and expresses electricity on either heating or cooling as charge accumulates, or as the lattice changes shape. The former is termed pyroelectricity, the latter piezoelectricity.”

Reviewer #2: Also, it is clear early on in the manuscript that the ice being studied is frozen water. It is not necessary to state “water ice” in the later sections of the manuscript.

Authors: Thank you for this suggestion. Twelve instances of “water ice” have been removed from places later in the manuscript. Sometimes “water ice” was retained and, this was done for clarity in some figures (which may be copied in the future by other authors) or in the text where specific mechanisms or properties of water ice are described.

Reviewer #2: The introduction should also explain why the fieldwork was undertaken in favor of a more controlled study in a laboratory setting.

Authors: Thank you for this good suggestion. The following text has been added as the second paragraph of the revised introduction to describe why this was done as field work and not in a laboratory setting.

“The present study was undertaken as field work. There was no clear model established yet. That arose in the failures and successes of the first experimental season, and inspired the experimental model that was followed during the second season with the experiment described in this paper. Real-world conditions and the various failures described herein and in the Supporting Information (S1 File) allowed for wildly disparate hypotheses to be worked on until a practical path became clear. Likewise, descriptions of technical problems and other errors that were eliminated during the present work are meant to help future research and to motivate more precise, controlled study in a laboratory setting as a follow-up.”

Reviewer #2: The Results section must also be greatly simplified. First, present experiments that worked, and the results and conclusions from these experiments clearly. Experiments that did not work should all be moved to Supplemental Information (section S3). The reviewer does not see the benefit of including the failed experiments in the main article. They make the key results difficult to comprehend.

Authors: This is excellent advice. All but the most successful experiment (Experiment 6) were moved to the Supporting Information file, for clarity and to improve the focus of the paper.

Reviewer #2: There are numerous tables and figures in the present version. Only representative data must be shown in the main article, and the rest moved to Supplemental Information.

Authors: Thank you for this insight. Yes, most of the tables and figures are now present only in the Supporting Information file. Yes, this is a good point.

Reviewer #2: The ordinate axis label for the power values in all the figures must be corrected, using the correct scientific format for numerical values.

Authors: Thank you so much for noticing this. I have added the following text to the figure caption to address this issue.

“Orders of magnitude for the power values are listed in-line for ease of reading but should be read as superscripts, e.g. the top left power value of the top left panel should be read as 10–8 W.”

Reviewer #2: While it is acknowledged that the authors are reporting only a preliminary study, a more detailed investigation of the electrical (ionic) conductivities of ice that contains the different additives would be desirable.

Authors: Thank you very much. The following text was added to the introduction to give the reader more details of the electronegativities of the monatomic ions and the dissociation coefficient of the dihydrogen phosphate.

“As with other solid electrochemical cells, the electronegativity of ions present in the structure of the ice helps to establish electromotive pathways and a voltage across the device. The monatomic ions studied in this experiment are K+, Na+, Cl– (with electronegativities: 0.82, 0.93, 3.16, respectively) and come from NaCl, HCl and KH2PO4 additives. [H2PO4]– has a dissociation coefficient of 7.20 and can both donate and receive hydrogen ions forming [HPO4]2– (dissociation coefficient 2.14) or H3PO4, phosphoric acid (dissociation coefficient 12.37).”

Reviewer #2: I recommend that the authors present a much more concise and to-the-point report of their study (consistent with the preliminary nature of the investigation) in a revised version of the manuscript.

Authors: Thank you for your time with this manuscript, and for this good suggestion. The introduction and results have been simplified. Most of the experimental data have been moved to the Supporting Information file with only Experiment 6 retained. The hypotheses have all been moved to the discussion section. The manuscript is now shorter, more focused, and clearer. Thank you so much for your time, insight and careful reading of the manuscript. The suggestions were very helpful.

---

## [Editor Report · Decision Letter 1]

25 Apr 2023

Electrochemical cells from water ice? Preliminary methods and results

PONE-D-23-02888R1

Dear Dr. Helman,

We’re pleased to inform you that your manuscript has been judged scientifically suitable for publication and will be formally accepted for publication once it meets all outstanding technical requirements.

Kind regards,

Nasser A. M. Barakat

Academic Editor

PLOS ONE
---

## [Editor Report · Acceptance letter]

12 May 2023

PONE-D-23-02888R1 

Electrochemical cells from water ice? Preliminary methods and results 

Dear Dr. Helman:

I'm pleased to inform you that your manuscript has been deemed suitable for publication in PLOS ONE. Congratulations! Your manuscript is now with our production department. 

Kind regards, 

on behalf of

Dr. Nasser A. M. Barakat 

Academic Editor

PLOS ONE